



# Quantifying multifrequency acoustic characterization accuracy for ice model development applications

**David R. Topham[1] and John. R. Marko[1]**

[1]ASL Environmental Sciences Inc., Saanichton, BC, Canada

5   **Correspondence:** David R. Topham (dtopham@aslenv.com)

**Abstract.**

Multi-frequency acoustic profiling is critically examined to estimate accuracies currently attainable in characterizing frazil suspensions: with primary interests focused on measuring fractional ice volume, a key factor in river ice growth models. The central issue is the adequacy of representations of backscatter cross sections of disk shaped 10   frazil particles in a well-established theory of elastic spherical targets.  An initial investigation established criteria for the existence of three-frequency solutions capable of providing lognormal statistical descriptions in terms of effective radii. These criteria restricted analyses of available field data with such models to inputs at two-frequencies limiting outputs to: a common effective radius, particle number density and frazil volume. Additional frazil cross section information is shown to be required to more fully exploit the full capability of multi-frequency profiling. An 15   approximate relationship between cross sections and the product of acoustic wavenumbers and particle effective radii ($k_1a_e$) is developed from laboratory polystyrene disk and sphere data and transformed into the natural ice environment.  Field data within the transformed range is transposed to higher frequencies in order to allow testing at still larger field values of $k_1a_e$.  Two-frequency analyses utilizing the resulting "pseudo-frazil" relationship confirmed a close match with the field data and increased compatibility with existence criteria for three-frequency 20   solutions.  The results showed that, within transducer calibration limits, the originally tested spherical backscattering extractions consistently under-estimate frazil ice volume concentrations by 25% confirming its continued use for accurate estimates in conjunction with a constant scaling factor of about 1.25.

## 1. Introduction.

25   Further development of numerical models for managing flow in a freezing river offers considerable potential benefits for public safety, hydroelectric production and proper functioning of cooling and drinking water intakes. However, accumulating evidence for simultaneous presences of multiple ice forms, such as suspended frazil, surface ice and in situ-grown anchor ice (Kempema and Ettema.2015, Jasek et al., 2015; Marko et al., 2015), is suggestive of serious, unaddressed, obstacles to model 30   improvement.  Progress has been constrained by a dearth of quantitative field data, particularly on sub-surface ice constituents. This situation was apparent in the recent work of Makkonen and Tikanmati (2018) which relied, almost exclusively, on laboratory results in calling attention to the practical importance of in situ anchor ice growth and needs for modelling it in the presence of other ice forms.

Data deficiencies are particularly evident in the frazil ice constituent. Its presence in the water column is 35   critically relevant to understanding the thermodynamics of overall ice production. Manual techniques for sampling frazil content, although widely employed in small streams (Dube et al., 2013), are impractical in large rivers.  Similarly, field applications of imaging at fixed points, developed for use in laboratory flumes and test tanks (Clark and Doering, 2006; McFarlane et al., 2015), have been restricted to shallow river edge locations (McFarlane et al., 2017). This technique is also intrinsically limited by reliance on 2-40   dimensional data to quantify 3-dimensional objects. Most applications have ignored unquantified volumes of irregularly-shaped particles: relying, instead, on probability statistics for diameters of particles deduced





by interpreting frazil images as projections of idealized circular disks. While contributing insights into frazil morphology, this approach has yielded little quantitative data on frazil volume, due to low data acquisition rates and the lack of capabilities for disk thickness measurements.

More recently, acoustic backscattering at multiple frequencies has been used (Marko et al., 2015) to study frazil in rivers during periods of ice growth and change. This methodology offers full depth seasonal monitoring of frazil production in large rivers at 1s or longer intervals: utilizing the fact that acoustic backscattering by frazil is largely a function of individual ice particle volumes and numbers. This sensitivity makes the technique ideally suited for measuring frazil fractional volume a key parameter in modelling the thermodynamics of ice growth. The multi-frequency technique is well established as a principal tool in quantitative particle suspension studies. Specific applications such as zooplankton monitoring (Chu et al., 1992), fish detection (Stanton and Chu, 2010), and sediment transport (Hay and Burling, 1982; Hay and Schaafsma, 1989) have been developed to high levels for research and monitoring purposes. Local measurements can be made simultaneously at high repetition rates throughout the water column from both self-contained and vessel-deployed instruments. Processing of detected acoustic signals is guided by the physical nature of the targets, and the information of interest.

Frazil studies have much in common with sediment transport monitoring. In both cases, extended time series are used to quantify volume fractions and suspended particle population statistics. Two important differences are that sediment particles are often, roughly, spherical in shape, with negative buoyancy (specific gravity 2.5), while frazil crystals (specific gravity 0.91) are slightly positively buoyant and, at low concentrations, predominantly disk-shaped. In modelling applications, where ice volume is of primary interest, the acoustic technique offers obvious advantages since near-neutral buoyancy renders the acoustic returns relatively insensitive to shape. Notwithstanding this feature, the extreme deviations of disk geometry from the spherical form, originally considered by Rayleigh (1897), have engendered some scepticism as to the accuracies attainable in interpreting the detected acoustic returns, particularly when the results strongly deviate from expectations. Key uncertainties reflect the inherent difficulties of verifying or calibrating scattering measurements on suspensions of otherwise "uncountable" numbers of fracturing, irregularly-shaped, melting and growing ice particles. Prior attempts to do this (Ghobrial et al., 2012, 2013) on laboratory frazil populations have not been successful (Marko and Topham, 2017).

Addressing this situation begins in the following section with an outline of backscattering measurement methodology which highlights features of the technology specifically relevant to calibration and verification. Three different approaches are taken in the subsequent evaluations. These include: examining the sensitivity of basic Rayleigh scattering theory to particle orientation and deviations from sphericity (Section 3); detailed laboratory comparisons of the scattering properties of disk- and spherically-shaped ice-surrogates (Section 4.1); and the use of field data to assess and improve internal consistency and interpretations of frazil backscattering at different acoustic frequencies (Section 4.2). The field data analyses draw upon and extend the laboratory-validated theoretical relationship tested in Section 4.1. A summary and conclusions are presented in Section 5.

## 2. Basic elements of acoustic profiling methodology

Extraction of information from particle suspensions by acoustic profiling requires detection of sound pulses backscattered by individual particles toward their original transceiver (transmitting and receiving) source. An essential element of extraction is that the detected fluxes of acoustic energy can be assumed to represent the arithmetic sums of fluxes arriving from all individual scattering particles: i.e. that the total cross section of all insonified particles is the sum of their individual cross sections. Modern instrumentation converts the received pulses into volume backscattering coefficients, $s_V(v_i)$, defined as the



scattered fraction of acoustic power at frequency, $v_i$, incident upon a unit volume of suspension which is detected by the transceiver. Knowledge of these coefficients at two or more acoustic frequencies allows quantitative descriptions of frazil in terms of particle population parameters. Doing this requires a valid theoretical relationship linking $s_V(v_i)$ to such parameters.

In general terms, the required relationship can be written for each of n different frequency channels as:

$$s_v^{theo}(v_i) = N \int_0^\infty g(a_e) \sigma_{BS}(a_e, v_i) da_e \qquad , \qquad (1)$$

where: N denotes the number of particles per unit volume; $\sigma_{BS}(a_e, v_i)$ is the theoretical backscattering cross section at an acoustic frequency, $v_i$, for a particle with an "effective radius" (defined below), $a_e$. The quantity $g(a_e)$, denotes a probability distribution for $a_e$ satisfying:

$$\int_0^\infty g(a_e) da_e = 1 \qquad . \qquad (1)$$

There are good theoretical reasons, confirmed by laboratory data (Clark and Doering, 2006; McFarlane et al., 2015)), for assuming that frazil sizes satisfy a two parameter lognormal distribution:

$$g(a_e, a_m, b) = \left[ (2\pi)^{0.5} b a_e \right]^{-1} e^{-0.5 \left( \frac{\ln(a_e/a_m)}{b} \right)^2} \qquad . \qquad (3)$$

The descriptive population parameters in this expression include the mean effective radius, $a_m$, and $b$, the standard deviation of the natural logarithm of $a_e$ which describes the "spread" in effective radius values.

Documenting a frazil population in such detail requires access to $s_V$ data measured at, at least, three frequencies. Estimates of $N$, $a_m$, and $b$ are obtained by minimizing a residual or quality parameter, $q$, defined as the sum over all channels of squared differences between measured and theoretical values of logarithmic backscattering coefficients:

$$q = \sum_{i=1}^{i=n} \left[ \langle S \rangle_v^{meas}(v_i) - S_v^{theo}(v_i) \right]^2 \qquad . \qquad (4)$$

The numbers of particles/unit volume within a range of effective radius, $da_e$, written as:

$$dN(a_e) = N g(a_e, a_m, b) da_e \qquad . \qquad (5)$$

allow fractional ice volumes, $F$, to be expressed in terms of optimized parameters as:

$$F = N \int_0^\infty \left( \frac{4\pi}{3} \right) a_e^3 g(a_e, a_m, b) da_e \qquad . \qquad (6)$$

The particle number density $N$, and the backscatter cross section, $\sigma_{BS}(a_e, v_i)$ are critical determinants of the accuracy of the derived physical properties; requiring both satisfaction of the independent scattering assumption and the availability of an appropriate backscattering model for use in Eq. 1. To simplify theoretical representations, disk-shaped particles, usually associated with frazil, are described in terms of
"effective spheres", characterized by an effective radius $a_e$, such that their volumes equal those of individual ice particles (Ashton, 1983). This simplification is facilitated the near neutral density of ice in fresh water. Measurements of $s_V(v_i)$ in, at least, three different frequency channels are required to derive the parameters $N$, $a_m$ and $b$ to describe the suspension. Given that quality acoustic field data essential for





the present work were confined to three frequency channels of the available data set, discussions are limited to considerations of two- and three-frequency extraction methods.

The less informative, but more easily applicable, two-frequency approach, previously used by Marko and Jasek (2010), limits the characterization of frazil suspensions to two parameters. In this case, Eq. (1) is

replaced by:

$$s_V(v_i) \quad = N^* \sigma_{bs}(a^*, v_i) \qquad , \qquad (7)$$

where N* represents per unit volume numbers of uniformly-sized spheres of radius, $a^*$, This representation is, of course, not realistically descriptive of a frazil suspension, but, instead, offers a convenient route to relatively robust estimates of a fractional volume parameter, $F^*$ , expressed as:

$$F^* = \left(\frac{4\pi}{3}\right) N^* a^{*3} \qquad . \qquad (8)$$

In general $a^*$ and $a_m$ differ, with $a^*$ exceeding $a_m$ ; a consequence of, overall, cross sections increasing with particle size, which dictates that more than 50% of the scattering by a lognormal frazil population involves larger than average particles. Representing such scattering levels in terms of a uniformly-sized population requires particles sizes larger than corresponding lognormal population means.

An unreliable data channel in the available 2011-2012 Peace River field program data limited analyses to

three of the four frequency channels. While applications of Eq. 4 still support extraction of optimal sets of frazil parameters ($N$, $a_m$ and $b$) and ($N^*$, $a^*$) from, respectively, the remaining three- and two-channel-based analysis, they represent exact solutions of the relevant equations. Solution can be simplified by expressing these equations in terms of ratios to eliminate the common factor of the particle number density $N$, ($N^*$). This step provides a set of equations which equates the ratios of the theoretical volume

backscatter coefficients in Eq.1 to their corresponding measured counterparts

$$G(i, j)^{meas} = G(i, j)^{Theo} \text{ for i, j = 1, 2, 3 with j not equal i} \quad . \qquad (9)$$

The terms in Eq. 9 are defined as:

$$G(i, j)^{Theo} = \frac{S_V^{Theo}(v_i)}{S_V^{Theo}(v_j)} \text{ and } G(i, j)^{meas} = \frac{S_V^{meas}(v_i)}{S_V^{meas}(v_j)} \quad .$$

The values of a* obtained in the two-channel solutions, and the values of $a_m$ and b produced for three

channels, are then substituted into Eqs. 7 and 1, respectively, providing two- and three-channel estimates of $N^*$ and $N$. The frazil fractional volumes, $F$ and $F^*$ are then calculated from Eq. 6 for or Eq. 8 for the respective three- and two-frequency extractions

For two frequencies this approach reduces the problem to solutions of a single equation in $a^*$. Three-frequency extraction necessitates simultaneous solution of two equations in $a_m$ and $b$, with the option of

three equivalent pairings of (i, j). All equations are solved numerically, re-casting Eq.9 in the least squares form of Eq. 4, to locate the zero of a residual, Q. This approach offers an advantage in that, in the event that no solution exists, a value of $a_m$, coupled with a vanishingly small value of $b$, is returned corresponding to a three-frequency-based optimization which closely approximates the mean value of the two-frequency solutions. This circumstance, indicated by $(1,3,4)_{b=o}$ is utilized further in Section 4.


The spherical target cross section relationships $\sigma_{BS}(a_e, v_i)$ required for these calculations are derived from updated (Anderson,1950: Faran,1951, Hickling, 1962) versions of the Rayleigh Theory of scattering by spherical targets (Rayleigh,1897). These relationships incorporate the elastic properties of the ice target material and, for brevity, will be referenced to their origins by the acronym "FEST", denoting Faran Effective Sphere Theory. The modern treatment of the algorithm is publically available as implemented by Dezhang Chu of the Northwest Fisheries Science Center (https://bitbucket.org/gjm/calibration-code/wiki/Home). Required inputs include: target radius, $a_e$ ; the ratios of target to fluid density ($\rho_2/\rho_1$); and wave speed ratios $c_{2s}/c_1$ and $c_{2L}/c_1$ (where $c_{2s}$ and $c_{2L}$ denote the shear and longitudinal sound speeds in the target (ice) and $c_1$ is the speed of sound in the fluid). The accuracy of this algorithm has been estimated (Dezhang Chu, personal comm.) to be +/- 0.001 dB or better for $k_1 a < 50$ where $k_1 = 2\pi/\lambda_1$ and $\lambda_1 = c_1/v_i$. This formulation is routinely used in calibrating precision sonar equipment with machined sphere targets.

Selected examples of the $G(i,j)$ ratio functions are shown in Figure 1 which maps the solutions in a graphical form for specific pairings of the measurement channels (i, j). It provides context for the following methodological descriptions. The frequencies are identified by data-specific channel numbers relevant to the field data of section 4.2.1, where the notation lists channels in descending order of acoustic frequency. In the figure, acoustic input data in the form of sᵥ ratios, $G(i,j)^{meas}$ are denoted by channel pair alone: i.e. (i,j).

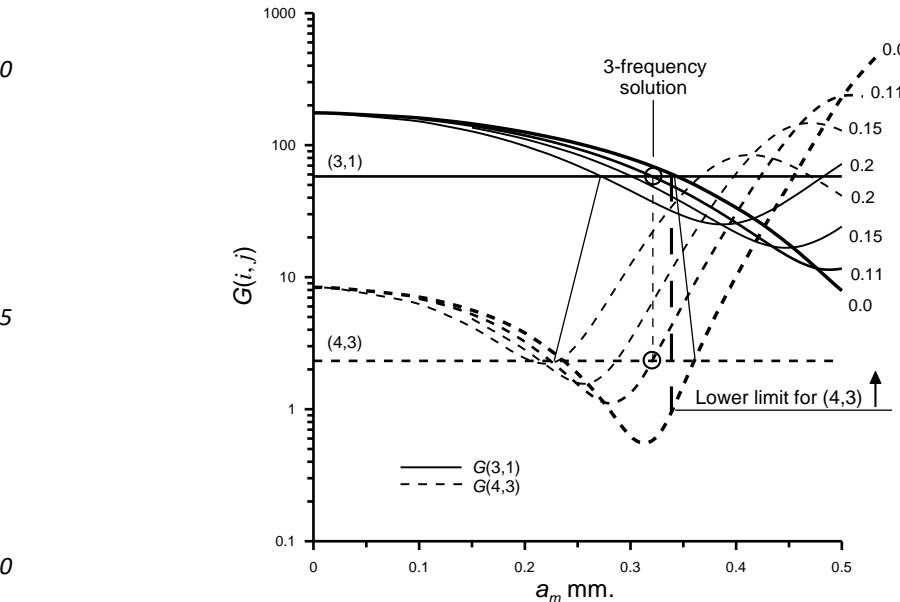

**Figure 1.** Illustration of relationships between two- frequency solutions in channels (4,3) and (3,1) to a three-frequency solution in channels (4,3,1).

The $G(i,j)^{theo}$ ratio for each pair is represented by a family of roughly parallel curves calculated by FEST as a function of radius, $a_m$, for the frequencies $v_i$ and $v_j$, with individual curves identified by the value of the lognormal distribution width parameter b. The minima in these curves define upper and lower branches for each plotted function. Each family is intersected by a straight horizontal line representing the corresponding $G(i,j)^{meas}$ ratio, specifically adjusted here to match the FEST model. The intersections



of this input data with the bold-lined b = 0 curves, taken individually, identify the two-channel solutions; the upper and lower branches of which provide a separate pair of valid a* and N* parameters. Resolution of this ambiguity requires information derived from conditions necessary for corresponding three-frequency solutions. The only requirement for the existence of the two-frequency solutions is that the input data lies within the bounds of the individual $G(i,j)^{Theo}$ curves.

The more complex three-frequency solutions require corresponding $G(i,j)^{Theo}$ curves in each of the two families to intersect their corresponding $G(i,j)^{meas}$ lines at identical values of $a_m$ and b. This necessity defines the appropriate branch of each function. In the example of Fig. 1, the $G(4,3)^{Theo}$ and $G(3,1)^{Theo}$ functions are paired through ,respectively, their upper and lower branches. As indicated above, the input data of $G(3,1)^{meas}$ = 58.0 and $G(4,3)^{meas}$ = 2.3, were pre-calculated with the FEST model to provide a known exact illustrative solution. The paired function intersection points of the solution, $a_m$ = 0.32 mm and b = 0.11, are marked by open circular symbols. The narrow sloping lines define the search area for the solution; the upper bound on b is imposed by the ability of $G(4,3)^{meas}$ to intersect $G(4,3)^{theo}$ . The lower boundary of search regions is established by the b = 0 $G(i,j)^{theo}$ curves. These limiting curves define initial input pairings of $a_m$ and b which lead to a valid solution; indicated on Fig 2b by the dotted line marked "initial value line". The radius $a_m$ is further constrained by the requirement that, for a given b, it not exceed the value associated with the intersection of the corresponding G(i,j) curves, at which point the relative positions of the functions are interchanged, leading to a spurious solution.

The necessary condition for the existence of a three-frequency solution defines a limiting relationship between the paired G(i,j) input values. For a given (3,1) data input, the $a_m$ value of the intersection with the $G(3,1)^{Theo}_{b=0}$ curve defines the point on the $G(4,3)^{Theo}_{b=0}$ curve corresponding to an exact solution, marked by a bold dashed vertical line. This point defines the minimum $G(4,3)^{meas}$ that can sustain a three-frequency solution for a given (3,1) channel pair. Lower values lie beyond the $G(4,3)^{theo}_{b=0}$ curve boundary, and therefore cannot match the $a_m$ value of the (3,1) pair. Satisfaction of this criterion to assure the existence of exact three-channel solutions can be established for a particular input data set prior to initiation of a search sequence. A mapping of the region associated with acceptable pairings of $G(i,j)^{meas}$ is included in Figure 2a. A further condition on the solutions is set by the upper bound on b, whereby the value of the $G(4,3)^{meas}$ input must exceed the minimum of $G(4,3)^{Theo}_b$. This requirement determines the maximum value of $b$ that can be accommodated by a specific value of $a_m$: i.e. the ($a_m$, $b$) values of the exact solutions must lie within the shaded area of Figure 2b. To realize exact solutions, the initial ($a_m$, $b$) input values must be below the dashed initial value

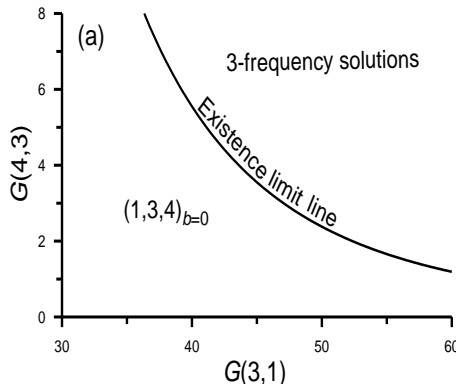

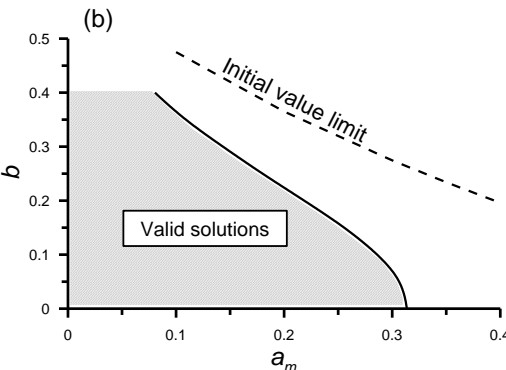

limit line.



**Figure 2.** (a) FEST existence limit line for three-frequency solutions. (b) Values ($a_m$, b) consistent with exact FEST-based solutions.

For input data positioned below the existence limit line of Fig 2a, searches for three-frequency solutions progress to points on the $G(i,j)_{b=0}$ curves. The corresponding residual $Q$ for such solutions reflects the proximity of the input $G(i,j)$ values to the existence limit line of Fig 2a. It will be demonstrated in Section 4.2.1 that this condition, denoted, by $(1,3,4)_{b=0}$, in the figure, yield estimates of fractional volumes which closely approoximate for the corresponding average of the (3,1), (4,3) and (4,1) two frequency fractional volume estimates.The branch pairing of the three-frequency solutions as b approaches zero necessarily defines the branch pairing for two-channel solutions: resolving the previously noted solution ambiguity.

**Table 1.** Branch pair combinations for 3-channel $b = 0$ and 2-channel solutions.

| Channel pair and branch combinations | |
|---|---|
| (4,1) Upper | (4,3) Upper |
| (3,1) Lower | (4,3) Upper |
| (3,1) Lower | (4,1) Upper |

The residual Q of the two and three-frequency exact solutions provides a measure of errors intrinsic to the least squares formulation, evaluated in the present work using a standard software package. For the two-frequency case, the residual for any identified solution is vanishingly small. For the more complex, three-frequency case, two outcomes are possible. These outcomes include the exact solution with vanishingly small residuals or, in the absence of such a solution, a $(1,3,4)_{b=0}$ result is returned with a larger ($Q$ on the order of 1) residual, which is, roughly, inversely proportional to the proximity of $G(i,j)^{meas}$ values to the existence limit line of Fig 2a. The small $Q$ value of an exact solution only signifies that it satisfies the set of existence conditions specified in Fig.2a with respect to input acoustic data. In this work, the bulk of the assessment process is based upon comparisons of estimates derived from measurements utilizing independent pairwise combinations of the three acoustic frequencies. The individual combinations span distinctly different portions of the range of the $k_{1s}a^*$ parameter which is the principal determinant in the critical cross section relationship. Differences in the obtained estimates signify the degree of mismatch between the acoustic model and data. Evidence supporting absolute validation of this relationship over a significant portion its range of variability is presented in Section 4.

### 3. Physical interpretations of acoustic backscattering by frazil suspensions

The foundations for practical applications of acoustic scattering were laid in Rayleigh's mathematical formulation of wave scattering by spherical targets. This work (Rayleigh, 1897) evolved into the Faran treatment of scattering by spherical elastic targets of radius a. For wavelengths long compared to characteristic body dimensions (i.e. for $k_1a \ll 1$), Rayleigh's treatment reduces to two leading terms of comparable magnitude: a monopole response to direct pressure changes; and a dipole contribution driven by changes in fluid velocity relative to the target. Both terms respond strongly to the volume of fluid displaced by the body which, in turn, experiences a reactive force in response to the acoustic wave. At this level of approximation, knowledge of the material properties and the inertia coefficient of the body are sufficient to define acoustic scattering properties. The long wavelength approximation provides useful representations of acoustic backscattering for $k_1a < 0.45$.

### 3.1 Single particle scattering.

The backscatter cross section, $\sigma_{BS}$, of a stationary obstacle of general shape can be expressed as a sum of monopole and dipole terms, constituting the Rayleigh long wave approximation, plus the sum of





remaining higher order terms represented by the symbol $O_n$. This explicit separation of the higher order terms is exploited in Section 4.2.2 to investigate their importance in the context of frazil ice suspensions. The full backscatter cross section expression can be written as:

$$\sigma_{BS} = \left(\frac{k_1^2 V_0}{4\pi}\right)^2 \left[\frac{K_2\text{-}K_1}{K_2} + (\Gamma\text{+}1) + O_n\right]^2 \qquad . \tag{10}$$

where $V_0$ is the volume of the obstacle; $k_1$ the incident wavenumber; and $K_1$ and $K_2$, respectively, represent the bulk moduli of the target and the fluid. $\Gamma$ is the inertia coefficient of the body, representing the rate of change in the kinetic energy of the velocity field associated with acceleration of the obstacle relative to the fluid. The characteristic length scale for the inertia coefficient is determined by the geometry of the induced flow field, rather than by the obstacle itself. For spheres, for example, the region of perturbed
fluid is closely associated with the target boundary and the characteristic length is the sphere radius. On the other hand, for a disk normal to the flow, the perturbed fluid occupies a roughly spherical region surrounding the disk, and the characteristic length is the disk radius. For a freely suspended obstacle, the dipole term is modified by the motion of the object in response to the reactive force of the acoustic wave. The momentum balance can be expressed in terms of the ratios:

$$\frac{U_2\text{-}U_1}{U_2} = \frac{\rho_1\text{-}\rho_2}{\rho_1 + \rho_2 \Gamma} \qquad . \tag{11}$$

where $U_2$ and $U_1$ are the velocity of the body and the fluid, respectively, and $\rho_2$ and $\rho_1$ represent corresponding densities. The dipole term of Eq. 10 can then be modified to give Rayleigh solution for a freely suspended particle.

$$\sigma_{BS} = \left(\frac{k_1^2 V_0}{4\pi}\right)^2 \left[\frac{K_2\text{-}K_1}{K_2} + \frac{(\rho_2\text{-}\rho_1)}{(\rho_2 + \Gamma\rho_1)} \cdot (\Gamma\text{+}1)\right]^2 \qquad . \tag{12}$$

The influence of the momentum balance on the dipole takes two distinct forms. The density difference between the fluid and the obstacle acts directly, while a more subtle interaction takes place between the
added inertia terms in the dipole itself. This feedback constrains the dipole within well-defined limits with magnitudes controlled by density differences.

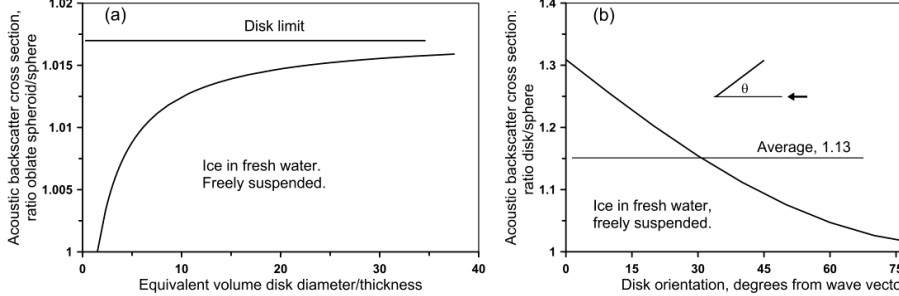

**Figure 3.** (a) Acoustic backscatter cross section for a freely suspended oblate ellipsoid relative to an equivalent volume sphere; (b) Effect of disk orientation on acoustic backscattering.

To chart the evolution of acoustic backscatter with changes in target shape, a freely suspended oblate ellipsoid of fixed volume was examined in the long wavelength limit as its geometry varies between the

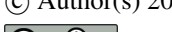

spherical and thin disk limits of the dimension along the symmetry axis. Our analyses were based upon a modified form of Rayleigh's treatment of the ellipsoid (Rayleigh 1897), to allow for free particle suspension. It can be shown that the backscatter cross section continues to scale directly with the parameter $(k_1^2 \, V_0/4\pi)$ through to the thin disk limit. The functional relationship itself depends on target geometry through the dipole term, but the overall scaling based on the "effective sphere" assumption continues to hold. For near-neutral density ice suspended in fresh water, the relationship remains very close to that of the sphere. Fig. 3a compares the scattering strength of an oblate spheroid of ice suspended in freshwater with that of a corresponding equivalent volume sphere as the length of the spheroid's axis of rotational symmetry is progressively reduced to the thin disk limit, with thickness defined to be that of a flat disk of the same volume. The insensitivity of the backscattering to shape details is evident.

### 3.2 Disk orientation.

For the disk, the strength of the dipole term varies as $\cos\vartheta$, where $\vartheta$ is the angle of the wave vector with respect to the plane of the disk. Since the dipole term is negative, the overall scattering decreases as the angle of inclination $\vartheta$ increases. Figure 3b shows the effect of orientation on a freely suspended disk compared to the sphere. For $\cos\vartheta = 0$, waves impinging the disk edge on, the dipole contributions vanish and the scattering is solely due to the monopole, increasing cross sections by about 30%. Such increases fall to about 2% as the disk becomes perpendicular to the wave vector. For a randomly oriented ensemble of disks, average backscattering strength exceeds that of the equivalent ensemble of spheres by about 12%, again through suppression of negative dipolar contributions from obliquely orientated disks.

The effective sphere concept as derived from the foregoing Rayleigh limit analysis encompasses two distinct concepts. Firstly, the "weak" effective sphere assumption offers an overall scaling parameter for the acoustic backscatter expressed as an "effective" spherical volume where, in general, the functional dependence of the backscatter is specific to the target geometry. The more restrictive, "strong" form of the assumption equates both the scaling parameter and the functional dependence of the scattering with those of spherical targets. In the case of the oblate ellipsoid, the volume-based scaling parameter remains valid over the full range of the transition from sphere to disk. The degree to which the strong form of the assumption is satisfied is determined by target and fluid density differences. For frazil ice in fresh water, near-neutral buoyancy suppresses geometrical effects, and the strong effective sphere requirement is almost fully satisfied in the Rayleigh regime.

Although the Rayleigh limit solutions are the dominant contribution to the scattering function at lower $k_1 a_e$ values, higher order terms, represented by $O_n$ in Eq. (10), become increasingly important above $k_1 a_e = 0.45$. The contributions of the latter terms for disk-like frazil targets are critical to the present investigation where both the strong and weak equivalent sphere assumptions play a central role in the interpretation of field data (Section 4).

### 4. Calibration and Verification

Validation of multi-frequency backscattering as a tool for characterizing frazil suspensions requires determining the degree to which its key assumptions of independent scattering and the effective sphere concept are satisfied in typical measurement environments. In prior applications to targets such as fish, zooplankton and sediments, validations frequently have been carried out on suspensions of surrogate target particles of known size, shape and volume concentration. The surrogates (acrylic and glass beads, polystyrene spheres, graphite- and stainless steel fluid-illed- shells (Stanton, 1988; Hay, 1991)), have, generally, differed in composition and detail from the targets of interest. Consequently, efforts tended to focus on verifying the quantitative relationships which link scattering cross sections to measurable physical properties of individual targets. For frazil particles, an alternative, more direct, approach,





utilizing scattering measurements on ice particle suspensions in a laboratory tank (Ghobrial et al., 2012, 2013), was frustrated by uncontrolled ice growth and the absence of credible independent validation methods (Marko and Topham, 2017). The present work takes a two–step approach which begins by reviewing and further interpreting laboratory surrogate data originally reported by Marko and Topham
(2015) as a basis for the subsequent step in which surrogate backscatter cross sections are transformed to the frazil environment and tested for consistency against field data.

The laboratory work, discussed in Section 4.1, used backscattering measurements at four different acoustic frequencies to quantify the relationship between backscattered returns and the compositions of particle suspensions in terms of numbers of individual particles and their physical and dimensional
properties. This critical step focused on establishing the accuracy of the weak effective sphere assumption for disk target suspensions, with $k_1a_e$ being the principal scaling parameter. It allows us to establish the degrees of error attributable to the limitations of FEST, enabling linkages of individual particle acoustic cross sections to key particle and measurement parameters over a limited, "validated", $k_1a_e$ range. This relationship provides the core of a methodology for using $s_v$ measurements in the ice/water system to
quantitatively characterize frazil populations. Section 4.2 progressively tests and refines this methodology on Peace River frazil data, utilizing the two- and three-channel processing approaches outlined in Section 2 to provide credible estimates of accuracies currently attainable in quantifying frazil content in natural river environments.

### 4.1 Laboratory measurements

The availability of precision-cut hexagonal polystyrene disks of a common thickness, and with a range of relevant diameters and effective radii allowed, Marko and Topham (2015) to calibrate mean individual cross sections in populations of identical surrogate targets with dimensions approximately duplicating those of circular disk-shaped frazil particles. Their results are briefly outlined, specifically, with respect to their relevance in supporting development of an equivalently well-founded relationship applicable to
measurements in the lower portion of the $k_1a_e$ range associated with typical river frazil measurements. Readers are referred to the original publication for fuller descriptions of methodology and less immediately relevant results.

Complications arose from polystyrene's major shortcoming as a frazil surrogate, which was that its characteristic shear wave speed is exceeded by the speed of sound in the host fluid (brine). This enables
polystyrene targets to sustain surface waves which are not accommodated by the Faran elastic sphere model, thereby introducing resonant transmission and scattering anomalies (Hay and Schaafsma,1989; Hefner and Marston, 2000, 2001) in polystyrene sphere and disk measurements.

**Table 2.** Polystyrene disk properties.

| Width (w) mm. | Thickness (t) mm. | Aspect Ratio (w/t) | Volume mm$^3$ | $a_e$ mm. |
|---|---|---|---|---|
| 0.38 | 0.125 | 3.05 | 0.016 | 0.155 |
| 0.5 | 0.125 | 4.06 | 0.028 | 0.188 |
| 0.81 | 0.125 | 6.5 | 0.072 | 0.258 |
| 1 | 0.125 | 7.92 | 0.106 | 0.294 |
| 1.6 | 0.125 | 12.8 | 0.277 | 0.405 |
| 2.39 | 0.125 | 19.1 | 0.617 | 0.528 |
| 3.18 | 0.125 | 25.4 | 1.091 | 0.639 |
| 6.35 | 0.125 | 50.8 | 4.365 | 1.014 |





The material properties of ice and freshwater preclude appearances of trapped surface waves in river frazil suspensions. Nevertheless, this phenomenon significantly affected development of methods for quantitatively interpreting frazil scattering data through its impacts on laboratory polystyrene in brine verifications. Specifically, avoidance of anomalies similar to those previously reported (Hay and Schaafsma, 1989; Hefner and Marston, 2000, 2001) required restricting laboratory measurements to situations where $k_1a_e \leq 0.58$. This restriction limited applications to, roughly, the lower half of the $k_1a_e$ range characteristic of natural frazil particle populations as typically studied at acoustic frequencies between 0.1 and 1.0 MHz. Compensating for this limitation in the following Section requires use of actual frazil field data but draws, critically, upon the validated portion of the laboratory results.

Those data were obtained from backscattering measurements made near-simultaneously at four different frequencies (125 kHz, 200 kHz, 455 kHz and 769 kHz) in a nearly cubic tank containing approximately 1 m$^3$ of brine titrated to assure neutral buoyancy for eight tested polystyrene ($\rho = 1056$ kgm$^{-3}$) disk species (Table 2) and a single species of microbead polystyrene spheres. The spheres were characterized by radius values, 0.295 mm, such that the volumes of the spheres closely matched those of the 1mm wide (effectively, the disk diameter) hexagonal disks: allowing direct testing of the effective sphere assumption as a function of acoustic frequency. This arrangement provided relatively stable, neutrally buoyant, suspensions with hourly concentration decay rates (due to settling and surface capture) low enough (between 5% and 8%) to allow averaging over 20 to 40 minute measurement intervals. The acoustic measurements utilized an ASL Environmental Acoustic Zooplankton Fish Profiler (AZFP) which was an updated, more efficient, version of the instrument utilized in the field studies discussed below.

To facilitate cross section estimates, two types of disk measurements were made: single species suspensions of progressively increased concentrations of identical particles; and mixed species tests. The procedures followed in these two cases differed in that single species concentrations were determined, as in the case of the microbead spheres, by suction sampling immediately before and after acoustic measurements, whilst, for practical reasons, mixture compositions were calculated from the weights of each added species, with each suspensions left undisturbed after an initial stirring. The results of single species tests for both disks and microbeads are shown in Figure 4 as normalized backscatter cross sections plotted vs. $k_1a_e$.

The microbead measurements provided an overall check on the experimental procedure and reference points for disk measurements. Comparisons with FEST calculations supported calibrations based upon polystyrene shear and longitudinal wave speeds of 1100 ms$^{-1}$ and 2380 ms$^{-1}$, respectively, chosen to optimally match measured and theoretical cross sections. These speeds differed only by 4% and 2%, respectively, from corresponding speeds calculated by Hay and Schaafsma (1989) from published elastic constants. The agreement achieved at the three highest frequencies, evident in the diamond–shaped data points, confirmed the basic reliability of the measurement technique and the applicability of FEST relationship with its predicted sharp local minimum at, $k_1a = 0.87$. Differences between the theoretical curve and values measured at the lowest, 125 kHz, frequency, denoted by the open diamond marker, were unexpected, given FEST robustness at low values of $k_1a$. These differences were suggestive of systematic measurement errors at 125 kHz. It was notable that the $k_1a$ values in the microbead measurements were well below the $k_1a \approx 1$ threshold identified by Hay and Schaafsma, (1989) for problematic surface wave resonances in polystyrene spheres.

All estimates of backscattering cross sections were limited to $s_v$ data showing the linear dependences upon concentration anticipated in Eq. 1. Linearity was restricted by discontinuities in slope for both disk and microbead species, marking the limit for the valid independent scattering assumption at a volume concentration of 0.07% identified by Marko and Topham (2015).

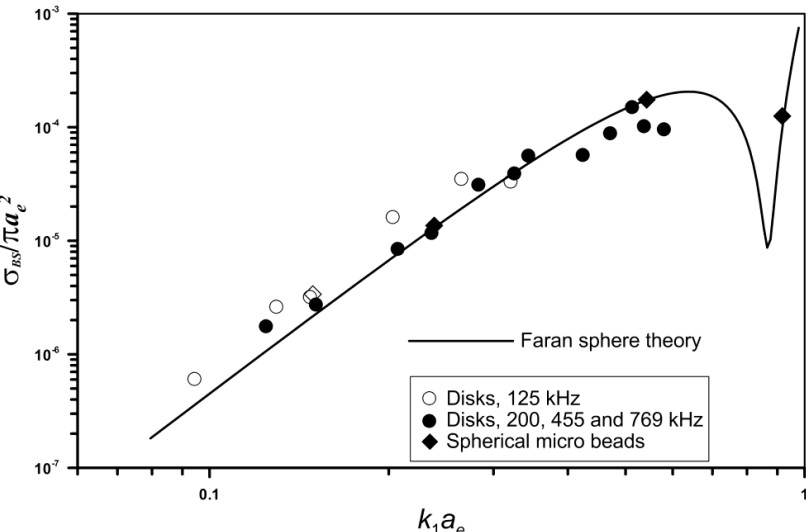

**Figure 4.** Summary of single species polystyrene disk and microbead normalized cross section results vs. $k_1a_e$. Disks and microbeads are denoted by circles and diamonds, respectively while; open markers
correspond to data acquired at 125 kHz.

Results of measurements on the hexagonal disks characterized in Table 2 included in Fig. 4 are restricted to values of $k_1a_e < 0.58$ to preclude surface wave generation. The 125 kHz results, including the microbeads, are denoted by open symbols and can be seen to be systematically elevated above the theoretical curve. This positioning is, again, in conflict with expectations that deviations from FEST curve
are negligible at very low $k_1a_e$ values and suggests that all 125 kHz data points shared a common measurement error. The remaining 200 kHz, 455 kHz and 769 kHz disk results (solid symbols) follow the theory but appeared to fall slightly below the corresponding curve beginning at $k_1a_e = 0.4$.

A second important part of the laboratory work was the mixed species testing which was designed to further quantify the independent scattering assumption in terms of the additivity of scattering
contributions from multiple individual target species. Inadvertently, these tests also provided definitive evidence for the systematic overestimation of single species 125 kHz cross sections. Mixture acoustic results were compared against two different expectations: firstly, from the sum of individual contributions as determined from the individual target backscattering coefficients measured in single species testing, and secondly, relative to a similar sum of contributions calculated using FEST theoretical cross sections.
The results fell into two groups distinguished by the response of the 125 kHz data relative to the alternative sets of expectations. For mixtures containing 0.38, 0.5, 0.8 and 1 mm width disks, and total fractional volumes rising progressively from 0.05 to 0.07, expectations based upon the single species measurements strongly overestimated mixture scattering, whereas FEST-based results, with the exception of mixture 1, achieved close agreement. This result mirrored the anomalously high 125 kHz scattering
levels identified in the single species tests depicted in Fig. 4. Observations later revealed long-lived turbulence structures introduced by the manual suction sampling, as the most likely source of these inconsistencies (see Marko and Topham (2015) for a fuller discussion of these errors). For mixtures with higher concentrations containing a small proportion of 1.6 mm particles, the overall concentrations rose to 0.15, considerably exceeding the non-linearity threshold of 0.06 obtained from the single species tests.



Primary interests in the present work are in the single species tests of the equal volume microbeads and 1 mm disks. These tests play a central role in the transfer of the measured backscatter cross sections into the natural frazil environment. The difference between the two cross section measurements can be used to eliminate the effects of measurement errors common to both target types such as cross overestimation at 125 kHz and transducer calibration uncertainties. When combined with allowances for the effects of neutral buoyancy, the differencing step facilitates the separation of the higher order terms of the disk response which is essential for transferring laboratory cross section information into the frazil environment.

**Table 3.** Comparisons between FEST and measured microbead and 1mm disk cross sections.

| Frequency kHz | $k_1 a_e$ | $\sigma_{bs}/\pi a^2$ Measured microbead | $\sigma_{bs}/\pi a_e^2$ Measured 1mm disk | $\sigma_{bs}/\pi a^2$ Faran theory microbead | Difference microbead-disk |
|---|---|---|---|---|---|
| 125 | 0.149 | 3.39E-06 | 3.21E-06 | 2.21E-06 | 1.82E-07 |
| 200 | 0.238 | 1.36E-05 | 1.18E-05 | 1.33E-05 | 1.88E-06 |
| 455 | 0.542 | 1.75E-04 | 1.02E-04 | 1.74E-04 | 7.29E-05 |

In summary, a very limited reworking of earlier work has identified abundant laboratory evidence that acoustic backscattering by polystyrene disk suspensions can be closely approximated for $k_1 a_e$ values < 0.58 in terms of scattering by volume equivalent spheres. Overestimation at the upper end of the allowable $k_1 a_e$ range was estimated to be on the order of 60% (2 dB). No correlations with disk aspect ratio were detectable within this range. The upper bound on $k_1 a_e$ was solely an artifact of the properties of the utilized surrogate material and host fluid: imposing no constraints on the accessibility of cross section information in the restricted $k_1 a_e$ range by other methods. Within the laboratory-validated $k_1 a_e$ range, a volume concentration limit of about 0.07% was established to avoid the breakdown of the independent scattering assumption at higher concentrations. This limitation also appeared to apply closely in mixtures of disk species, implying the existence of similar limits on frazil measurements.

### 4.2 Verification by Field Data.

Section 4.1 offers strong evidence that FEST quantitatively represents backscattering by polystyrene disk suspensions in terms of effective spheres when $k_1 a_e$ values are less than 0.58 and volume concentrations do not exceed 0.07%. The limitation on $k_1 a_e$ was specific to the polystyrene/brine system and its capability to support surface wave generation. Such waves, although excluded from FEST, introduce discontinuities into the acoustic results at above-threshold $k_1 a_e$ values. No similar limitations were anticipated in frazil measurements. Moreover, the results of Section 3 showed that, in the long wavelength Rayleigh limit, scattering by thin disks suspended in freshwater can be closely represented in terms of the effective sphere concept and spherical scattering theory. At the $k_1 a_e = 0.58$ limit of laboratory validations, however, polystyrene disk cross-sections were roughly 2 dB below those measured for spherical particles of equal volume. This change was attributed to differences between the higher order terms (in $k_1 a_e$) of spherical scattering theory and their equivalents for disk-shaped targets.

These differences are explored below by applications of two complementary approaches to data acquired in the same Peace River field program which provided the basis for the original Marko et al. (2015) frazil analyses. The first approach, described in 4.2.1, involves direct applications of the strong effective sphere assumption embodied in FEST. It evaluates the consistency of frazil fractional volumes estimated using different combinations of acoustic frequencies. The weak equivalent sphere assumption is invoked in Section 4.2.2 to adapt and apply laboratory polystyrene disk results to the frazil environment. This work establishes the extent to which the assumed $k_1 a_e$ dependences of frazil disk cross sections are compatible



with the foregoing sensitivity results and the full body of multifrequency acoustic field data. In both cases, the accuracies of frazil characterization, particularly in terms of fractional volume, require transfers of understandings between two different target and host fluid systems. The details of this process and, specifically, its implications for the accuracies of multifrequency frazil fractional volume estimates are discussed. All results draw upon the flexibility and redundancy offered by multifrequency measurements.

Field measurements utilized a four frequency Shallow Water Ice Profiling Sonar (SWIPS) unit (manufactured by ASL Environmental Sciences Inc.) operating at 125 kHz, 235 kHz, 455 kHz and 774 kHz. The transducers for three channels (channels 1 (125 kHz), 3 (455 kHz) and 4 (774 kHz)) were mounted in a common moulded head and integrated into a pressure case separated by 30 cm from an isolated 235 kHz (channel 2) transducer. All instruments, including an ADCP (Acoustic Doppler Current Profiler) were mounted in a weighted instrument package deployed on the riverbed about 25 m from the riverbank in 5 to 6 m water depths. Electric heating discouraged acoustic beam blockage by anchor ice accretion.

Frazil characterization parameters were extracted with customized RUNSWIPS software based upon the strong scattering assumption and FEST. This software included capbilities for limited testing of deviations from a fully FEST-based treatment of backscattering using the multiple two-channel processing procedures outlined in Section 2. In principle, the extractions could have utilized four different combinations of three frequencies and six combinations of two frequencies. However, problematic instabilities in the 235 kHz channel 2 output (likely from debris accumulation) precluded its use. This difficulty limited analyses to data acquired with three combinations of two channels (channels 3,1; 4,1 and 4,3) and a single triplet of frequencies (4,3,1) (Marko et al., 2015, 2017; Marko and Topham (companion paper), 2020).

### 4.2.1 Sensitivities of FEST-based interpretations of frazil data to $k_1a_e$

Analyses were carried on 2-minute averaged $s_v$ data acquired from levels 2.6 m above the SWIPS instrument during a representative 8-hour Mar. 20 Peace River freezing interval selected to be associated with a stable and representative body of frazil-related acoustic returns. This work quantified the sensitivities of frazil characterizations to measurements which included $k_1a_e$ values exceeding the 0.58 limit of laboratory validations. These comparisons assumed full FEST applicability, temporarily ignoring the small overestimates of cross sections at the upper end of the laboratory-validated $k_1a_e$ e range. Focus was given, to the sensitivities of $F(t)$ and other outputs to the $k_1a_e$ parameter. Evidence was sought for major inconsistencies potentially introduced by FEST in applications to non-spherical particles. Similar analyses of a Mar. 22 interval yielded nearly identical results which are not presented here.

Initial analysis drew upon results obtained from data acquired simultaneously in two different frequency channels, enabling direct comparisons of measurements made in different ranges of the $k_1a_e$ parameter. Time series of $F^*$, $N^*$ and $a^*$ (two-channel- equivalents of $F$, $N$. $b$ and $a_m$) for the three available channel pairings: (3,1) (4,1) and ((4,3) were calculated for each averaging interval. Plots of $F^*(t)$ in Fig. 5a-c showed very similar time dependences. Initial quantitative comparisons focused on results contemporary with peak fractional volumes as estimated at times centered around 100 minutes into the interval. Key results and relevant parameters deduced for this period are summarized in the first three rows of Table 4 for all channel pairings. The entries in the first column denote pair composition and the relevant branch (Table 1) of the corresponding $G(i,j)$ curve required for a valid set of frazil parameters. The subsequent columns in each row present outputs for $a^*$, $F^*$, and $N^*$, together with ratios of peak $F^*$ values relative to the mean of the three alternative two-channel estimates. The final two columns in the Table list values of $k_1a^*$ corresponding to the lowest and highest acoustic frequency associated with each extraction. The





final Table row includes the means of the three two-channel results. The full mean two-channel $F^*(t)$ time series is compared in Fig. 5d with a curve representing the outputs of three-channel, $(1,3,4)_{b=0}$, extractions. The latter comparison reinforces the conclusion, cited in Section 2, that such extractions, consequences of failed attempts to identify true lognormal size distributions, effectively represent the

averages of all available two-channel extraction estimates.

Overall, the estimates of $a^*$ from the three different pairings of channel data are grouped closely around a mean value of 0.36 mm. More significant deviations from the mean are apparent in the $N^*$ results and in the, closely related, fractional volume estimates. From the compilation in Table 4, differences between peak $F^*(t)$ values estimated using, alternatively, the (3,1) and (4,3) inputs are approximately 10%. This

agreement was surprisingly good given that, on the basis of the $k_1a_e$ limits listed in the last two columns of Table 4, approximately 75% and 25% of the, $k_1a_e$ ranges spanned by, respectively, the (3,1) and (4,3) measurements were within the validated portion of the $k_1a_e$ regime. In other words, measurements in the two pairs of channels were, respectively, well- and poorly-validated. On the other hand, the factor of 1.55 discrepancy between the mean of these two estimates and those obtained with the (4,1) pair was

worrisome. While inconsequential relative to the nearly two orders of magnitude differences separating modelled and measured frazil contents (Marko et al.,2015), the observed difference was suggestive of unexplained channel sensitivities. Given the character and generally high quality of the available data, the possible presence of such sensitivities justified further analyses to identify their origins and, more generally, to further enhance the accuracy of multi-frequency profiling. .

**Table 4.** Results of three different two-channel extractions applied to $s_v$ data at the peaks of the $F^*(t)$ and $F(t)$ curves plotted in Fig. 9a-d.

| Channels | Peak $a^*$ | Peak $F^*(t)$ | Peak $N^*(t)$ | $F^*(t)/{}^*F_{Mean}$ | Lowest utilized value of $k_1a^*_e$ or $k_1a_m$ | Highest utilized value of $k_1a^*_e$ or $k_1a_m$ |
|---|---|---|---|---|---|---|
| 3,1 (lower) | 0.404 | $3.51 \times 10^{-5}$ | $1.27 \times 10^5$ | 0.888 | 0.227 | 0.826 |
| 4,3 (upper) | 0.363 | $3.08 \times 10^{-5}$ | $1.54 \times 10^5$ | 0.781 | 0.770 | 1.309 |
| 4,1 (upper) | 0.350 | $5.29 \times 10^{-5}$ | $2.96 \times 10^5$ | 1.331 | 0.196 | 1.217 |
| Means | 0.373 | $3.96 \times 10^{-5}$ | $1.92 \times 10^5$ | 1.000 | 0.212 | 1.310 |

One obvious objective for such analyses would be to assess the impacts of the, roughly, 2 dB differences between FEST- and laboratory measured-cross sections at the upper end of the validated $k_1a_e$ regime. It

was recognized that this and other analytical efforts were hindered by the fact that $s_V$ data inputs consistently failed to satisfy the existence criteria identified in Section 2 for obtaining three-frequency solutions based upon the FEST cross section relationship. Consequently, at least initial assessments had to be confined to two-frequency analyses with $F^*$ as the primary output. Ultimately, improvements in the utilized cross section relationship are needed to allow optimal fitting of three-channel $s_v$ data to produce

realistic logarithmic effective radius distributions capable of raising levels of confidence in two-frequency produced $F^*(t)$ outputs.

To assess the likelihood of such improvements, it is instructive to examine the effects of changes in $b$, the width of the particle size distribution, on fractional volumes and mean effective radii extracted with the three-channel approach. This effort involved runs of the RUNSWIPS processing algorithm for

progressively larger, fixed, values of $b$ on $s_V$ input data corresponding to both the peak and flat sections of the Mar. 20 $F^*(t)$ curves plotted in Fig. 5a-c. Very similar values of $a_m$ and $F/F_{b=0}$ were obtained





throughout the frazil interval. Values of these quantities, plotted in Fig. 6a, corresponding to the peak portions of the test interval, show $a_m$ decreasing and $F/F_{b=0}$ rising smoothly from unity as $b$ increases from zero. The third curve in the Figure depicts the sensitivities of corresponding $q$ values as calculated from Eq. (4). Effective radius distributions corresponding to the configurations: $b = 0$, $a_m = 0.36$ mm and

$b = 0.18$, $a_m = 0.28$ mm, compatible with these curves, are included in Fig. 6b as representative of, respectively, uniformly-sized and lognormal frazil distributions which differ in fractional volume by roughly 30%.The lognormal parameters $b$ and $a_m$ in the latter case are, roughly, compatible with recently reported field-estimates (McFarlane et al., 2017), of equivalent frazil disk diameter distributions, assuming the 15:1 disk diameter to thickness ratio suggested by flume data (Clark and Doering,

2006). The factor of two elevation in the $q$ values for the selected lognormal parameter configuration relative to the $b = 0$ distribution is within the range of uncertainties introduced by the utilized FEST cross section relationship. These results support the conclusion that the absence of optimal three-channel solutions for non-zero $b$ values was a likely consequence of small errors in the assumed FEST cross section relationship. In principle, this absence can be addressed by improving the cross section

relationship and, possibly, by increasing the accuracies of $s_V$ measurements.

The data points in Fig. 6a were derived from limited numbers of measurements made in the (Marko et al., 2015) study during a highly dynamic January 14 interval. The underlying $s_V$ data, in this case, were distinguished by being compatible with the existence criteria identified in Section 2 for three-channel solutions. The extracted results were indicative of relatively low $b$ values and comparatively large mean

effective radii. The accompanying fractional volume estimates can be seen to have been scaled up from their corresponding uniform-size values in accord with the curve developed from the Mar. 20 data. The close match of their absolute $a_m$ values to the simulated increases is largely coincidental and indicative of the small range of variation in the entire Peace River frazil data set (Marko et al, 2015). Nevertheless, these results further raise prospects for using improved representations of the scattering process  to

narrow current extraction uncertainties beyond those inferred from two-channel analyzed results. This work, described in Section 4.2.2, is focused upon expanding the validated range of the $\sigma_{BS}(k_1 a_e)$ cross section relationship.





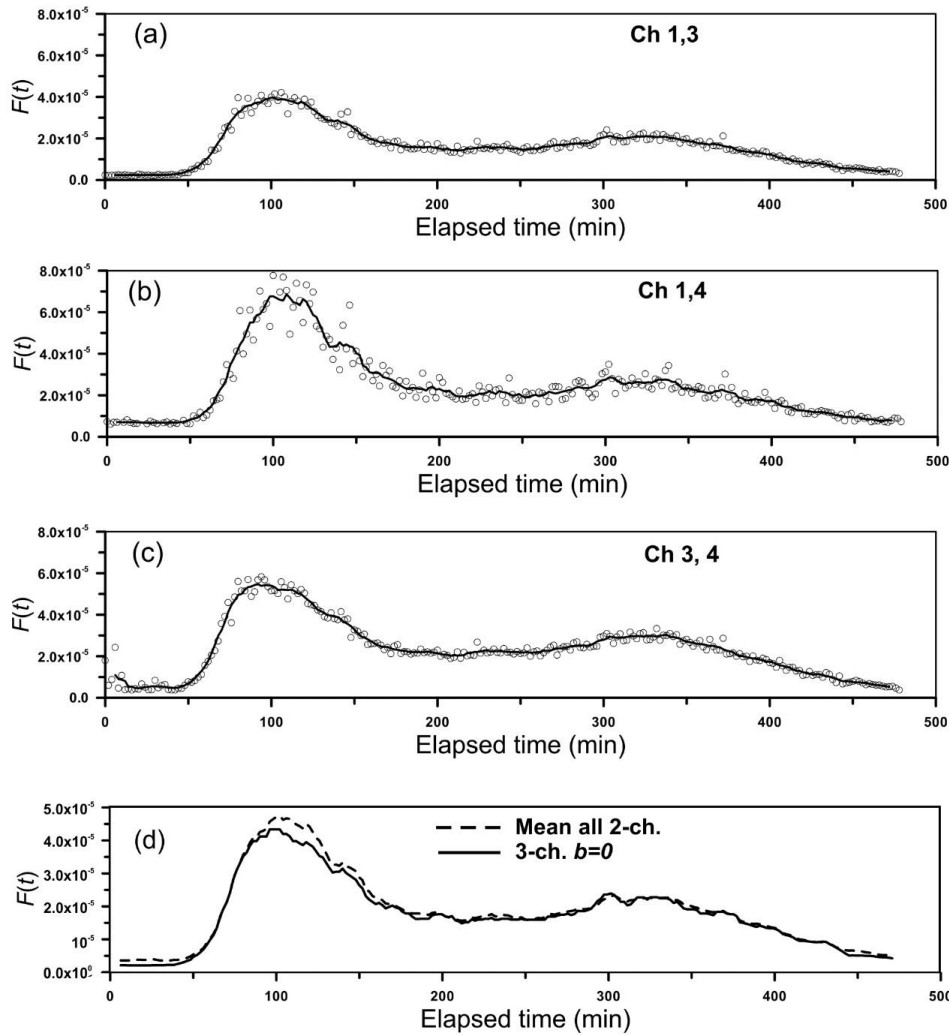

**Figure 5.** Comparisons of the March 20 frazil fractional volume as derived with the channel combinations: (a) (4,3);(b) (3,1);. (c) (4,1) and (d) comparison of means of three-channel results as derived with all channels (i.e. (4,3,1) assuming b =0)

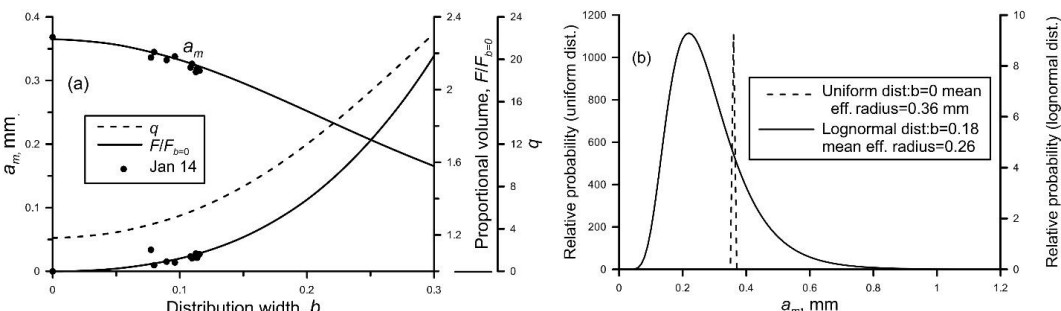

**Figure 6.** a) Plots of mean effective radius, $a_m$ , and q  as derived using three-channel (4,3,1) extraction a function of b ; and b) Plots of relative effective radius probability for:   =0, $a^*$=0.36 mm; $b = 0.18$  and $a_m$ = 0.26 mm .

### 4.2.2 Frazil cross section validation at $k_1a_e$ values above the laboratory-validated range

The laboratory studies of scattering by polystyrene disks and spheres outlined in Section 4.1 showed disk backscattering cross sections departing measureably from  FEST expectations as $k_1a_e$ values rose beyond 0.4. These differences, attributed to higher order terms, were approximately, -2.4 dB at $k_1a_e = 0.58$, the limit of valid measurements on polystyrene disks.  While allowing frazil fractional volume estimates with, roughly, +/-50% accuracies in the lower half of the typical range of $k_1a_e$, variability, field data suggested significant limitations were introduced by full reliance on an elastic sphere model. Further advances require both additional transfers of limited laboratory verification results into the freshwater frazil environment and corresponding extensions to higher $k_1a_e$ values. These steps required quantifying the role of higher order scattering contributions and their sensitivity to target shape.

Work to that end was initiated with explorations of the general characteristics of the higher order terms in existing spherical target scattering models.  It was found to be convenient to represent normalized cross-sections in the generalized format of Eq. 13, which assumes effective sphere scaling in which $M$ and $D$ and $O_n$ respectively, represent monopole, dipole and higher order term contributions.

$$\frac{\sigma_{BS}}{\pi a_e^2} = \frac{(k_1 a_e)^4}{9\pi} \left[(M + D) - O_n\right]^2$$       (13)

The effect of the higher order terms is to reduce the Rayleigh scattering. (The negative sign of $O_n$ is introduced in Eq. 13 to facilitate logarithmic presentation.) The long wavelength Rayleigh contribution (introduced in Eq. 10 of Section 3) can be written in the same format as:

$$\left(\frac{\sigma_{BS}}{\pi a_e^2}\right)_{Ry}^{1/2} = \frac{(k_1 a_e)^2}{3\sqrt{\pi}} \left[(M+D)\right]$$       (14)

This allows the higher order terms to be isolated and expressed as:

$$\left(\frac{(k_1 a_e)^2}{3\sqrt{\pi}}\right) O_n = \left(\frac{\sigma_{BS}}{\pi a_e^2}\right)_{Ry}^{1/2} - \left(\frac{\sigma_{BS}}{\pi a_e^2}\right)^{1/2}$$       (15)





The terms on the right hand side of Eq. (15) can be determined from routine backscatter cross section calculations which apply FEST to a neutral buoyancy polystyrene sphere in brine, and an ice sphere in freshwater. Additionally, to illustrate behaviour in the absence of shear forces, the Anderson (1950) model was applied to liquid spheres with bulk moduli and densities equal to those of ice, The normalized theoretical backscatter cross sections, $\sigma_{BS}/\pi a_e^2$, for each of these possibilities are plotted in Fig.7a as functions of $k_1 a_e$.

The most prominent features of the two elastic sphere cross section curves (A and B) are the distinctive local minima in the vicinity of $k_1 a_e = 1$, which vary dramatically in depth and width, reflecting differences in host fluid and target material properties. The critical parameter distinguishing the plotted curves was the ratio of shear wave speed in the target material to the speed of sound in the host fluid. For polystyrene/brine, characterized by the most pronounced minimum, this ratio is less than unity (which, as noted previously, restricted the applicability of laboratory data). In the ice/freshwater case the shear wave ratio exceeds unity, giving rise to a correspondingly less prominent curve minimum. The shear-free "liquid ice" sphere curve, did not exhibit a minimum.

The corresponding sums of higher order terms, $\left(\frac{(k_1 a_e)^2}{3\sqrt{\pi}}\right) O_n \left(\frac{(k_1 a_e)^2}{3\sqrt{\pi}}\right) O_n$ are plotted in Fig. 7b, where the ordinates of the ice and liquid ice sphere curves are shifted upwards by multiplication by factors of 10 and 100, respectively, to facilitate viewing individual curves. The higher order terms are remarkably similar for all three sphere models and, to a large extent, obey simple power laws (bold solid lines), For the two elastic sphere models, these power laws extend beyond $k_1 a_e$ values associated with the peaks of the cross-section curves (denoted, in Fig.7b, by adjacent arrows). Additional calculations (not shown) for ice spheres in neutral density suspensions showed the effects of density differences to be largely confined to the dipole terms, leaving higher order terms unaffected.

Interpretation of multi-frequency frazil data rests critically on the laboratory cross section results reported in Section 4.1 for the matched volume, 0.295 mm radius, polystyrene spheres and 1mm wide hexagonal disks. These data provided a direct test of the effective sphere assumption. The matched volumes ensure the equality of corresponding monopole terms, and, since the neutral density of the suspensions eliminates dipolar contributions, comparisons of the two sets of scattering data give direct measures of higher order term differences. Subtraction of this difference (Table 3, Section 4.1) from the matching Faran solutions then provides error-corrected backscatter cross sections for disks at $k_1 a_e$ values of 0.149, 0.238 and 0.542, plotted in Fig. 7a. Use of Eq. (15) with these data points then provides estimates of the higher order terms of the corrected disk cross sections, which also follow a simple power law, depicted by the dotted line in Fig.7b. Combined with Eq.13, this power law can represent the 1 mm polystyrene disk backscatter cross section relationship as denoted by the dotted line curve (D) in Fig. 7a.



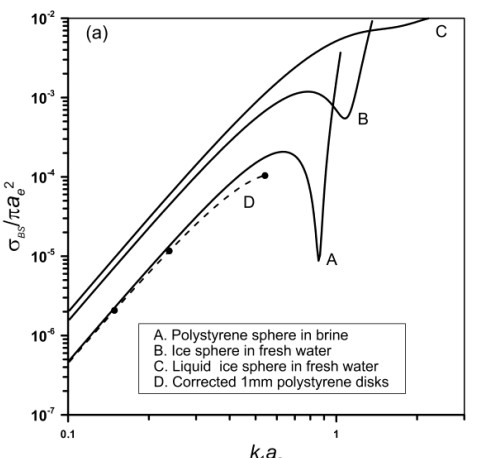
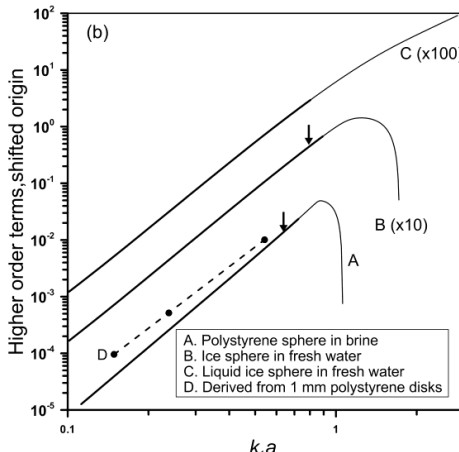

**Figure 7**. (a) Backscattering cross section of relevant varieties of spherical and disk-shaped polystyrene and ice targets. (b). Higher order terms extracted from these cross sections, as described in text. Origins of curves 2 and 3 are shifted by multiplication by factors of 10 and 100 for clarity purposes: Arrows denote the positions of corresponding peaks in (a).

To apply these results to frazil ice disks, cross sections deduced from the polystyrene/brine measurements must be transformed into equivalent quantities in the ice/freshwater environment. The near equality of the slopes of polystyrene/brine and ice/water higher order terms Fig.7b, suggests the feasibility of replacing the sum of higher order terms in the FEST ice/freshwater solution by an appropriately-scaled multiple of the corresponding polystyrene disk sum. To confirm this equivalence, the transformation was first applied to the ice and polystyrene sphere solutions using power law approximations which, as indicated in Figure 7b, remain valid for $k_1a_e$ values approaching the peaks of corresponding cross section curves. The polystyrene/brine Rayleigh components translate directly into ice/water equivalents upon insertion of appropriate material parameters. The higher order ice/freshwater terms represented by $O_n$ are replaced by the polystyrene/brine power law after multiplication by a factor of 1.37 to align the frazil peak with the peak of the ice/freshwater sphere. The transformed curve, marked by the dotted line B is almost indistinguishable from the original sphere solution at below-peak $k_1a_e$ values.

To transform randomly oriented, neutrally buoyant, polystyrene disks, the Rayleigh solution is replaced by the equivalent oblate spheroid solution for randomly oriented ice disks (Section 3), together with higher order ice disk terms derived from the corresponding polystyrene disk power law after multiplication by same 1.37 factor applied to the spherical case. The original (A) and transformed (C) disk solutions are represented by the short dash-line curves in Fig. 8. The peaks beyond the validation limits are a natural consequence of summing the Rayleigh 4th order power law and the power law approximation of the higher order terms. Although such peaks are observed in the original sphere model, in the absence of experimental confirmation, their presence in disk suspensions is speculative but not unexpected. The 2.4 dB difference between cross sections of equal volume polystyrene spheres and disks in neutral density brine (A) near the peak in Fig. 8 is attributed the higher order terms. This difference translates into a 1.4 dB difference in the frazil disk case (C); where random orientation reduces the negative dipolar contributions of buoyant disks.

Transferring the validation limit on polystyrene disks to frazil applications is unavoidably subjective in the absence of a fixed relationship between "equivalent" $k_1a_e$ values in the two measurement





environments. Taking the peaks of the extrapolated disk curves as a reference, a linear transfer of $k_1a_e$ values associated with the polystyrene/brine validation limit to the frazil disk curve places the equivalent frazil validation boundary near $k_1a_e = 0.7$. For convenience, the transformed disk curve will henceforth be referred to as the "pseudo-frazil" backscatter cross section curve. The possible existence of a peak suggests that frazil disk cross sections may be relatively insensitive to small extensions beyond the validated region.

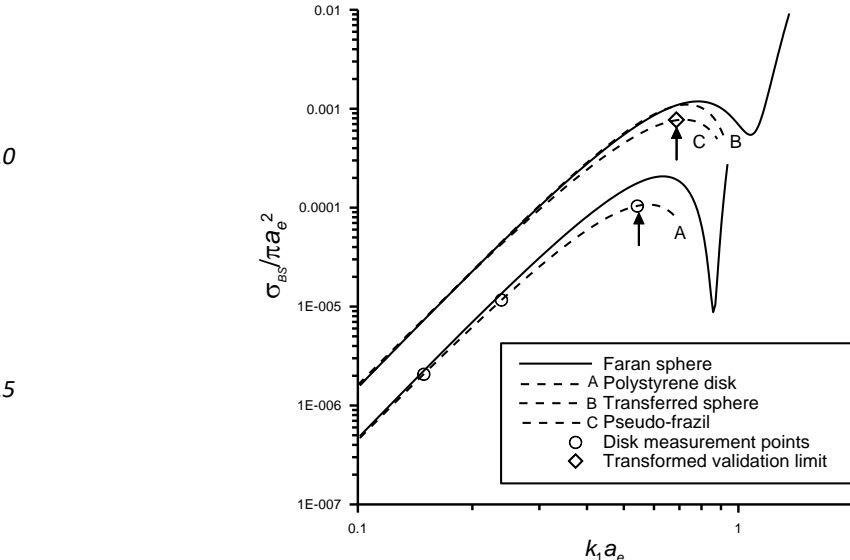

**Figure 8.** Modifications of FEST cross section vs $k_1a_e$ relationships to accommodate laboratory polystyrene disk in brine results and the scaled transfer of higher order dependences between polystyrene /brine and ice/freshwater systems. The short dashed curves, in each case, represent the best available estimates of the validated measurement regimes on disk shaped targets. In each case, arrows denote the confirmed upper bound of validated measurements.

The data in Table 4 show field data being acquired at $k_1a_{e\,e}$ values > 1.0, well outside the validated range. Moreover, the inferred relatively flat, or slightly down-trending shape of the frazil disk curve immediately beyond the validation limit, offers little guidance on cross sections in the higher $k_1a_e$ regime. Consequently, additional information was required to realize even a coarse extension of the pseudo-frazil curve.

Access to such information was available through additional utilization of the multi-frequency character of the Peace River field data with the aid of a simple "bootstrap" technique. This approach uses frazil population parameters, as derived within the validated region, to impose constraints allowing $s_v$ data, simultaneously acquired in still higher frequency channels, to provide estimates of $\sigma_{BS}$ at $k_1a_e$ values, which, because of the higher frequency, were outside of the validated regime. These estimates serve a function similar to the laboratory experiments of Section 4.1, in which backscattering data acquired on suspensions of identical, well characterized, targets of known concentration, provided estimates of mean individual target cross sections. In this case, two-channel processing is used to derive values of $N^*$ and $a^*$







from data acquired at the two frequencies compatible with $k_1a_e$ values in the validated regime. The resulting value of $N^*$ is used in Eq. 16:

$$s_V(v_i) \quad = N^*\sigma_{BS}(a^*,v_i) \qquad . \qquad (16)$$

to estimate $\sigma_{BS}$ at $k_1a_e = k_1a^*$ with $k_1$ representing a higher acoustic frequency associated with $s_v$ measurements outside of the validated regime. Similar extensions can be obtained from all unique

pairings of channels within the validated $k_1a_e$ regime. The acquired quantities, $\sigma_{BS}(k_1a^*)$, are cross sections associated with a population of uniformly-sized frazil particles, characterized by an effective radius a*. Except, perhaps, for extremely broad distributions of particle sizes ($b > 0.3$) this procedure yields cross section estimates compatible with those derived from laboratory measurements on single species sphere and disk populations. In the absence of channel 2 in the Peace River data set, the only

feasible bootstrap option in our study employed two-channel extractions from channels 1 and 3 data, followed by use of Eq. (15), in conjunction with contemporary channel 4 $s_v$ data. The Mar. 20 frazil interval, again, provided stable, smoothly changing, $s_V$ values, indicative of a set of relatively time independent $a^*$ values. Given the restriction to three frequencies, bootstrap transfer of information was confined to a very small range of $k_1a_e$ values compatible with stable $s_v$ values over typical, few minute-

long, averaging periods. In short, the FEST based analysis showed a narrow range of $a^*$ values to be characteristic of the entire data set: indicating that the $k_1a_e$ range spanned by bootstrap outputs tends to be largely determined by strategic choices of channel numbers and frequencies. This is evident in Fig. 9a where the added estimates of normalized cross sections occupy a narrow range of $k_1a_e$ near $k_1a_e \approx 1.2$: falling, roughly, 2dB below expectations indicated by the FEST theoretical curve. Cross section points at

the upper end of validated region, and as derived from the channel 4 bootstrap procedure are included in Fig. 9a in normalized form.

The pseudo-frazil curve of Fig.8 has been extended in Fig.9a to include the "bootstrap"-derived cross section estimates by a simple polynomial, empirically adjusted such that the resulting channel $\sigma_{BS}$ difference ratios $G(i,j)$ enclose the corresponding $s_V$ ratios indicated by the field data. This requirement,

graphically represented in Fig. 9b by shaded rectangles marking the maxima and minima of the respective $s_V$ ratios, ensures the existence of, at least, 2-frequency solutions for this data set. The range of validated 2-frequency solutions is limited to values of $a^*$ below about 0.5 mm by the $k_1a^* \approx 1.3$ limit associated with the bootstrap-generated points in Fig 9a.

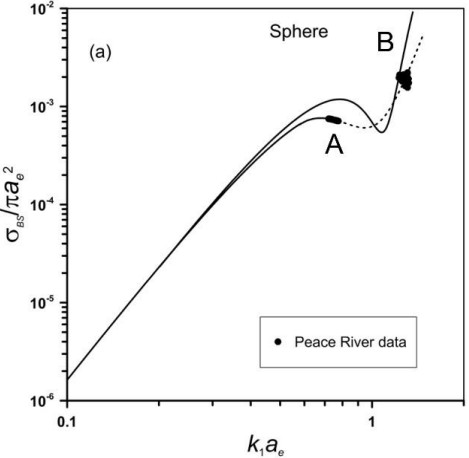

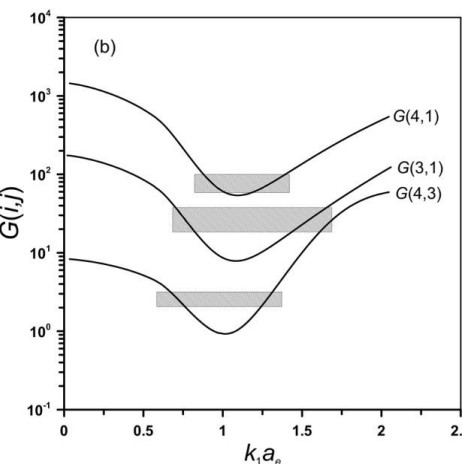





**Figure 9.** (a) Extended pseudo-frazil normalized cross section (A), compared to FEST values as calculated for spherical ice particles (B). (b) Plots of viable $G(i,j)$ ratios, the height of the shaded rectangles demarcate the ranges of compatible the ratios of $s_v$ values observed in corresponding constituent channels during the Mar. 20 interval.

Within the limitations of available field data, the extended pseudo-frazil curve in Fig. 9a offers an initial step toward a reliable model for interpreting backscatter from natural frazil suspensions. The principal differences between this model (A), and the FEST based relationship (B), are the 1-2 dB overestimates introduced in the latter case for $k_1a_e$ values between 0.6 and 1.25. These differences, for typical Peace River frazil particle sizes, primarily impact upon channel 3 and channel 4 measurements and are likely

sources of inconsistency in the two-channel FEST-processed $F*(t)$ results listed in Table 4. Clearly, given the paucity of in-hand data currently available for comparisons, full replacement of the FEST cross section relationship and, ultimately, reliable extraction of all parameters descriptive of a lognormally distributed frazil population, must remain an aspirational objective: requiring additional acoustic measurements. On the other hand, as suggested in Section 4.2.1, limited cross section adjustments such as

those effected by the pseudo-frazil curve, could have prospects for raising capabilities for estimating, at least, the critical fractional volume parameter to levels compatible with needs for supporting model refinements and understanding physical processes. Some encouragement for this optimistic view was detected in the fact that a significant proportion of input data from the tested Mar. 20 interval satisfied solution existence criteria for pseudo-frazil-based three-frequency processing.

To explore such possibilities, the data underlying the two-channel FEST-based results of Table 4 were reprocessed with the pseudo-frazil curve of Fig. 9a. The results, listed In Table 5, showed that pair to pair differences in estimates of $a*$ tended to be compensated by corresponding, opposing sign, changes in $N*$ which maintained frazil content close to a common level. This robustness of $F*$ with respect to the utilized range $k_1a_e$ values is, again, evidence of the fundamental role played by ice volume in determining

the strength of acoustic scattering. Importantly, the individual pair estimates of $F*$ all fell within 15% of their mean value, considerably improving upon the 30% spread of the corresponding FEST-processed results of Table 4. The enhanced consistency increases confidence in the more refined measurement technique which also raised peak Mar. 20 $F*(t)$ values about 30% above FEST-based estimates.

**Table 5**. Average channel differences from data points at elapsed times of 88, 94 and 110 min into mar.
20 interval as processed with alternative combinations of two- channels and assuming cross sections given by the pseudo-frazil curve in Fig. 9a.

| Channels | Peak $a*$ | $F*(t)$ | $N*(t)$ | $F*/F*_{Mean}$ | Lowest utilized value of $k_1a*$ | Highest utilized value of $k_1a*$ |
|---|---|---|---|---|---|---|
| 3,1 (lower) | 0.359 | $4.84 \times 10^{-05}$ | $7.49 \times 10^5$ | 1.040 | 0.584 | 0.732 |
| 4,3 (upper) | 0.372 | $5.12 \times 10^{-05}$ | $7.16 \times 10^5$ | 1.100 | 0.758 | 1.290 |
| 4,1 (upper) | 0.388 | $4.00 \times 10^{-05}$ | $5.03 \times 10^5$ | 0.860 | 0.476 | 1.346 |
| Mean | 0.373 | $4.65 \times 10^{-05}$ | $6.56 \times 10^5$ | 1.000 | 0.560 | 1.294 |

The presence of significant scatter in the (4,1)-derived $F*(t)$ data (Fig. 5b) at times close to the peak in the Mar. 20 frazil content record necessitated some efforts at quality control underlying analyses leading

to the results presented in Tables 4 and 5 These efforts limited extractions to just three 2 minute averaged points corresponding to times 88, 94, and 110 minutes after the start of studied frazil interval. All these points satisfied the criteria, outlined in Section 2, for extraction of optimal three-frequency frazil characterizations based upon, in this case, the extended pseudo-frazil cross section relationship. Similar


$F^*(t)$ values were obtained at all three points with each processing approach, in spite of the widths $b$ of the corresponding lognormal distributions varying between 0.05 and 0.17, The individual ratios of the pseudo-frazil- to FEST-based $F^*(t)$ estimates varied by less than 1% from an average values of 1.17.

Focusing on data compatible with optimal three-channel pseudo-frazil-based extractions allowed more detailed and broader examination of $F^*$ sensitivities to the assumed cross section relationship. Specifically, alternative two-channel analyses methods were applied to two small sets of data selected to be associated with contrasting $b$ and $F^*$ values: with each set comprised of just two similar time series points. The first set, with mean values of $b$ and $F^*$ of 0.029 and $2.0 \times 10^{-5}$ respectively, from about 390 minutes into the time series, provides a strong contrast with the second; selected from the peak region with values of $b = 0.15$ and $F^* = 4. \times 10^{-5}$. The resulting FEST- and pseudo-frazil-based estimates of F* are listed in Table 6 as the mean estimates in each ($b$, $F^*$) category after averaging all three two-channel pairing possibilities

**Table 6.** Comparisons fractional volume results from two selected Mar. 20 2-minute averaging periods associated distinctly different values and processed, alternatively, with the pseudo-frazil and FEST cross section relationships.

| Processing | $b$ | $F^*_{Mean}$ | $F^*/F^*_{Mean}$ Pair (3,1) | $F^*/F^*_{Mean}$ Pair (4,3) | $F^*/F^*_{Mean}$ Pair (4,1) | $a^*_{Mean}$ | FEST Correction ratio |
|---|---|---|---|---|---|---|---|
| pseudo-frazil | 0.029 | $1.86 \times 10^{-5}$ | 1.005 | 1.010 | 0.985 | 0.360 | 1.19 |
| " | 0.150 | $4.45 \times 10^{-5}$ | 1.055 | 1.132 | 0.813 | 0.369 | 1.18 |
| FEST | 0.029 | $1.56 \times 10^{-5}$ | 0.870 | 0.762 | 1.368 | 0.369 | |
| " | 0.150 | $3.76 \times 10^{-5}$ | 0.906 | 0.821 | 1.273 | 0.370 | |

The values of $b$ played no part in the analyses other than to test the extent to which the uniform particle assumption, intrinsic to the two-frequency approach, alters $F^*$ estimates. For the narrow distribution associated with $b = 0.029$, the two-frequency analysis provides a realistic description of the suspension, and the comparison between the two processing methods largely reflects the differences in the cross sections assumption. The low $b$ pseudo-frazil analysis showed channel pair differences below 2%, suggestive of the absence of significant differences in the pseudo-frazil curve of Fig. 9a and the actual cross section relationship. Corresponding FEST results, with channel pair differences approaching 40%, were consistent with the presence of significant errors in the assumed cross section as discussed above. At the higher values of $b$, the effective radius output of pseudo-frazil analysis rose slightly, as did the accompanying channel pair differences: consistent with the limitations of two-frequency analysis. The FEST results, on the other hand, are more erratic, showing the largest channel pair differences at the lowest $b$ value.

Although the two samples chosen for analysis have very different particle distribution widths and mean volume concentrations, the values obtained with the pseudo-frazil processing exceeded FEST results by an almost constant ratio, in spite of the large differences in the consistency of corresponding individual channel pair estimates. The resulting ratios of the two $F^*$ means, listed in the final column as "FEST correction ratios", were almost identical to those associated with the data in Tables 4 and 5: again attesting to the internal consistency of the input data and interpretations

These results, albeit deduced from a small set of samples, support adoption of the present pseudo-frazil cross section model as a basis for the more informative frazil characterizations available from three channel extractions. Accordingly, two-channel comparisons based upon this model were extended



without restriction to a cover the full durations of the Mar. 20 Interval. These results along with an independent set of set of pseudo-frazil-based extractions are presented in Figs. 10a and 10b, respectively. The included data points encompass $b$ values which decrease from a maximum of roughly 0.2 at the beginning of the frazil interval, falling close to a 0.06 minimum at the peak before rising again during the initial decay period. In both Figures, FEST two-channel means are represented by the 7 point weighted mean curve of Fig 5d. Figure 10a compares the mean values of estimates derived from data collected by channel pairs (3,1), (4,3) and (4,1) subjected to, alternatively, FEST- and pseudo-frazil-based analyses. A companion set of exact 3-channel solutions is compared with the latter curve in Fig. 10b. It is to be noted that the $F^*(t)$ and $F(t)$ comparisons utilize different vertical scales for representations of, respectively, FEST- and pseudo-frazil-based results: specifically adjusted to maximize the visual correlation between the two time series.

The scale adjustment factor of 1.2 of Fig. 14a for two-frequency processing matches results of Table 4 and 5, together with the more selective data of Table 6. Any effects of particle distribution width are too small to be distinguished within the scatter of the data. The only discernable difference between the three-channel results of Fig. 10b relative to Figure 10a was the presence of some evidence for a marginally larger scaling factor of 1.3. Statistically, the pseudo-frazil $F^*(t)$ and $F(t)$ results were essentially interchangeable. In summary, it appears that a simple scaling of FEST-based 2-channel analysis by a mean factor of about 1.25 largely reproduces $F^*(t)$ results obtained when the extended pseudo-frazil cross section relationship is used for two-channel processing. $F(t)$ estimates are also feasible using the three-channel processing methodology without additional scaling provided that a basic relationship linking cross sections to $k_1 a_e$ is available to allow he avaialble $s_V$ data to satisfy solution existence criteria.

Confident and complete three-channel-derived descriptions of frazil populations in terms of volume particle number densities and lognormal distribution parameters, almost certainly require further field or laboratory measurements to extend and complete the currently interpolated section of the pseudo-frazil curve of Fig. 9a. Fractional volume estimates, on the other hand, are clearly currently obtainable from two-channel extractions based upon data acquired at three or more frequencies, and processed with either the pseudo-frazil cross section relationship, or FEST-based extractions upwardly scaled by a factor of 1.25. In the latter case, it is convenient to take advantage of the effective equality, demonstrated in Fig.5d, between fractional volumes estimates as derived from the three-channel, $(1,3,4)_{b=0}$, extractions and two-channel estimates after averaging over all three possible pairings of measurement channels. In all cases, systematic uncertainties of +/- 30 %, normally associated with transducer calibrations, can be assumed to be built into otherwise unbiased, roughly +/15%, estimates of ice content.

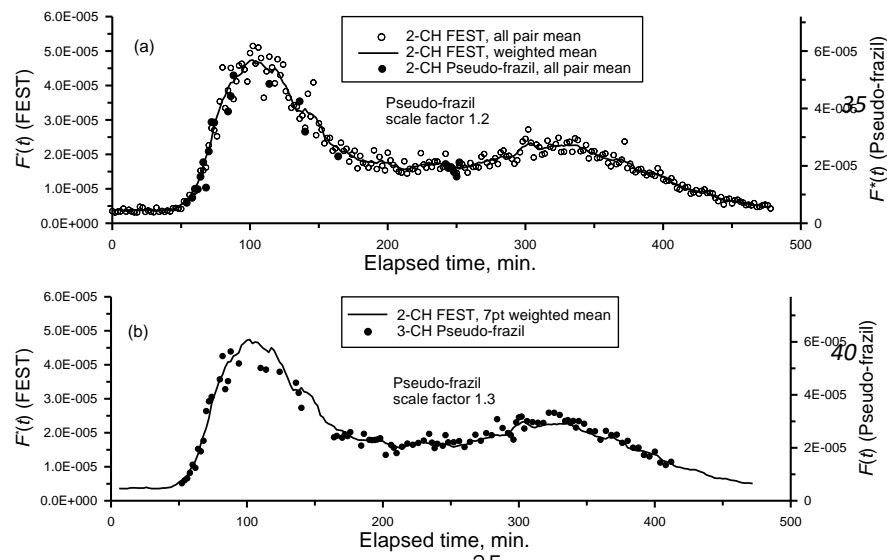



**Figure 10.** Comparison of pseudo-frazil and FEST based processing: a) 2-channel pair means; b) FEST 2-channel pair mean with 3-channel pseudo-frazil. The vertical scale of the pseudo-frazil data is adjusted to maximize the visual correlations.

## 5. Summary and conclusions

Building on basic theoretical concepts, new analyses of laboratory and field data assessed the accuracies currently attainable in characterizations of suspended frazil with multi-frequency acoustic backscattering techniques. Principal interests lay in quantifying the errors introduced into estimates of water column frazil content from use of the effective sphere approximation and an elastic sphere scattering theory to interpret acoustic data. The approach examined the mathematical foundations of the utilized extraction procedures to clarify relationships between the two- and three-frequency-channel processing procedures used in earlier work. The results highlighted the sensitivity of the more detailed three-channel outputs to currently imperfect knowledge of theoretical backscatter cross sections and errors in $s_v$ measurements. A Mar. 20, 2012 Peace River frazil interval provided a suitable data set for analysis and assessment.

The validation procedures were critically anchored to earlier laboratory studies which established the quantitative accuracy of a spherical scattering model for frazil surrogate suspensions of polystyrene disks for $k_1a_e$ parameters in the lower half of the range encountered in field measurements. The 0.58 upper $k_1a_e$ limit was imposed on the laboratory verifications by the properties of the surrogate material. At this limit, the utilized FEST (Faran Effective Sphere Theory) spherical scattering model over-estimated cross sections of disks by about 2 dB. Initial evaluations of errors at larger $k_1a_e$ values utilized comparisons of frazil fractional volume estimates derived with alternative pairings drawn from data collected in three functioning acoustic frequency channels. One of these pairings spanned almost the full laboratory-verified range, while the second and third pairings, encompassed the upper half and nearly the full entirety, respectively, of the field-encountered range. Agreement was achieved, to within, approximately, 10%, between estimates based upon, alternatively, the first and second of these pairings (corresponding, respectively, to the lower (verified) and upper (unverified) halves of the field data range). This result was taken as indicative of the absence of major errors in the utilized FEST-based approach. However, the, roughly, 50% larger fractional volume estimates attained with the third measurement pair left open possibilities that measurement uncertainties could reach several tens of per cent. Consequently, additional work was undertaken to further reduce uncertainties in estimates of $F^*(t)$ and, eventually, to facilitate access to particle distribution statistics through extensions of the laboratory disk results to encompass a larger fraction of frazil field measurement conditions.

To this end, higher order contributions to cross sections calculated from established spherical scattering models and equal volume spherical and disk-shaped frazil surrogates were merged and scaled to provide a validated $k_1a_e$ range relevant to natural frazil environments. This transformation produced a pseudo-frazil backscatter cross-section relationship which could be validated to $k_1a_e$ values approaching 0.7. Bootstrap methods which combine two-channel extractions from the two lowest frequency channels with $s_v$ data acquired in the highest frequency channel, provided critical independent cross section estimates in a narrow range near $k_1a_e = 1.2$. When combined with the previously validated regime, the resulting



estimates allowed use of standard curve fitting techniques to produce an extended pseudo-frazil cross section curve as a credible, if still imperfect, alternative to the FEST relationship utilized in earlier two- and three-channel extractions. The modified relationship offered two immediate benefits. Firstly, it brought all three different two-channel peak fractional volume estimates to within 10% of their mean value which was about 20% larger than the corresponding mean of the more erratic estimates obtained with FEST-based processing. Secondly, the altered cross section dependence on $k_1a_e$ greatly increased the fraction of the two-minute-averaged time series $s_v$ data points satisfying requirements for valid three-channel extraction. This change opened possibilities for more detailed extractions of additional parameters; potentially allowing full descriptions of frazil suspensions in terms of particles with lognormally distributed effective radii.

The absence of a fourth frequency precluded fuller applications of the bootstrap extension process in the present study. Nevertheless, the results were sufficient to both demonstrate the utility of the extension technique and clearly established current capabilities for estimating frazil fractional volumes. In our view, the identified changes in the FEST $\sigma_{BS}(k_1a_e)$ relationship were not sufficiently comprehensive to justify immediate replacement of the FEST-based automated RUNSWIPS processing algorithm. Such a step, while ultimately desirable, will require measurements at additional acoustic frequencies to obtain bootstrap cross section estimates in several other narrow ranges of the $k_1a_e$ variable. One obvious option would be to add measurement capabilities at frequencies roughly in the middle of the gaps separating the three frequencies utilized in the present work. Data acquired at one such frequency, 235 kHz, were, in fact, acquired during the Peace River studies but were not of usable quality. A successful deployment at this frequency and in the vicinity of 600 kHz would contribute cross section estimates in three additional narrow $k_1a_e$ ranges outside of the laboratory-validated regime. The additional channels would also allow multiple alternative modes of three-channel extraction: enabling direct comparisons of independent characterizations to further narrow the uncertainties in estimates of fractional volumes and other frazil population parameters.

At present, however, the results strongly suggest that measures of fractional volumes to absolute accuracies approaching the +/-30% systematic limitation intrinsic to transducer calibrations are obtainable from FEST-based processing of multi-frequency data when used in conjunction a 1.25 multiplication factor. The quality of such estimates should be more than sufficient to resolve outstanding issues on the respective roles of frazil and anchor ice in river surface ice formation. One of those issues, the impacts of *in situ*-grown anchor ice on frazil growth, first raised to explain the Marko et al. (2015) results, has been partially clarified by recent confirmations (Kalke et al., 2015; Evans et al., 2017; McFarlane et al., 2017; Makkonen and Tikanmati, 2018) of significant anchor ice presence in freezing rivers. Nevertheless the volumes of such ice, the circumstances governing its growth, and its connections to frazil ice content remain largely unquantified. The resulting uncertainties, which considerably inhibit effective modelling, are addressed in the following paper through analyses of SWIPS acoustic data acquired during early winter portions of the 2011-2012 Peace River study.

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
