# Peer review of "Quantifying multifrequency acoustic characterization accuracy for ice model development applications"

_The Cryosphere, 2020_

## Referee Comment (RC1) · Anonymous Referee #1 · 30 Oct 2020

General comments:

I do not think the majority of this manuscript will be of interest to the readers of Cryosphere; its content is too theoretical. Moreover, in its current form the manuscript meanders to much from theory to lab, back to theory, to field, and so on. The manuscript reads more like a diary of the trials and tribulations of the research undertaken by Topham and Marko over a number of years rather than a focussed manuscript. That said, I think the paper contains some very useful results.

I am not a theoretician. I study river ice freeze-up and frazil events.

The manuscript looks at the theoretical work on using multi-frequency transducers (at

least 3) to determine the number of particles per unit volume, N, mean effective radius, a_m, and standard deviation, b. Previous work has used 2 frequencies, which yields N* and a*, the numbers particles per unit volume, and the uniformly-sized radius of the spheres, respectively. The analysis relies on the work of Rayleigh (1897) and Faran (1951) to provide the theoretical foundation for the acoustic scattering. Polystyrene spheres and disks provide the acoustic scattering in the laboratory while frazil ice (and whatever else might have been in suspension) provides the acoustic scattering in the field.

Polystyrene spheres and disks have different acoustic scatter properties – and both of these behave differently than frazil ice, which while small are largely disks. I don't think the Cryosphere audience needs 27 single spaced pages to compare and calibrate these scatters. Even the summary and conclusions are too verbose.

I would recommend a major revision focussed at the Cryosphere audience.

I have no specific comments given I believe the manuscript needs a major distillation for this journal's audience.

---

## Referee Comment (RC2) · Anonymous Referee #2 · 7 Nov 2020

This study looks at acoustic quantification of the volume of suspended frazil ice in a riverine context. It does this by combining laboratory measurements with in situ observations through application of a backscatter model.

Major The structure is awkward and makes it difficult to work out what has actually been done. There are quite a few pages of introductory material but then it looks like a theoretical model has been developed. Is it novel?

The manuscript results in a quite compact and quantitative conclusion. It would seem sensible to actually foreshadow this in the Introduction to aid in focus.

Section 3 on physical interpretation – I couldn't work out if this was introduction, meth-

ods or Discussion? At the very least a clear identification of what is novel to this manuscript for the analysis is required.

Section 4 – so this is where the new work is described? But then quite a bit of text seems still introductory?

Referring the reader to the previous paper is ok but there needs to be a minimum of information here. As it is I eventually found it in on Page 11. So the data in Figure 4 are new?

The systematic difference of Fig 4 for the 125 kHz data is clear. Is there an indication of uncertainty in the estimates and is it independent of frequency. This gets discussed and reference made to a previous paper for a fuller discussion. Can they at least be represented on the Figure?

The field data used for validation have been previously published. This is OK as it is being used to evaluate a "new" model. But quite specific points are made about the data with no context provided here. What's a "representative 8-hour Mar. 20 Peace River freezing interval"? (pg 14/25). Or "a highly dynamic January 14 interval"?

Possibly a naïve question but are there any data with which to compare the estimates of Fig 5? As it is these are combinations of the acoustically sensed data and it is hard to make sense of it when it is all self-comparison.

The start of the Summary (pg 26/10) is succinct and clear and I suggest a version of this would be very handy as say the second paragraph of the Introduction after the wider context.

Minor There are many wayward commas that are either not necessary or in the wrong place.

Pg 1/25 The opening sentence could be improved. Is it the numerical models or actually improved understanding that we really need?

Equations a mix of italics and not italics.

It would be useful to cite some ocean literature for example the recent Frazer et al. 2020 in GRL https://doi.org/10.1029/2020GL0904989oi0